# Predictors of Unmet Needs in Chilean Older People with Dependency: A Cross-Sectional Study

**DOI:** 10.3390/ijerph20206928

**Published:** 2023-10-16

**Authors:** Raffaela Carvacho, Marcela Carrasco, María Beatriz Fernández, Claudia Miranda-Castillo

**Affiliations:** 1Millennium Institute for Care Research (MICARE), Santiago 8370146, Chile; raffaela.carvacho@micare.cl (R.C.); mrfernan@uc.cl (M.B.F.); 2Faculty of Medicine, Pontificia Universidad Católica de Chile, Santiago 8331150, Chile; 3Center for Studies in Age and Aging (Centro de Estudios de Vejez y Envejecimiento CEVE-UC), Pontificia Universidad Católica de Chile, Santiago 7820436, Chile; 4Institute of Sociology, Faculty of Social Sciences, Pontificia Universidad Católica de Chile, Santiago 7820436, Chile; 5Faculty of Nursing, Universidad Andres Bello, Santiago 8370146, Chile; 6Millennium Institute for Research in Depression and Personality, Santiago 7820436, Chile

**Keywords:** needs assessment, Camberwell Assessment of Need for the Elderly, mental health, older people, functional dependency

## Abstract

Approximately one in five Chilean older adults has some degree of dependency. Limited evidence is available on self-perceived needs in Latin-American older people. The main aim of this study was to identify predictors of unmet needs of dependent older persons without cognitive impairment, considering personal and primary informal caregivers’ factors. This cross-sectional study was conducted with a sample of 77 dyads of older people with dependency and their caregivers. A survey was administered, evaluating sociodemographic characteristics, anxious and depressive symptomatology, health-related quality of life, and social support. Older people’s self-reported met and unmet needs and caregivers’ burden and self-efficacy were also assessed. To determine predictors of unmet needs, a multiple regression analysis was carried out. Most participants had mild to moderate levels of dependency. The most frequent unmet needs were “daytime activities” (33.8%), “company” (23.4%), “benefits” (23.4%), and “psychological distress” (24.7%). Older people’s higher level of dependency and anxious symptomatology were predictors of a higher number of unmet needs, with a model whose predictive value was 31%. The high prevalence of anxious symptomatology and its relationship with the presence of unmet needs highlight the importance of making older people’s psychological and social needs visible and addressing them promptly.

## 1. Introduction

The accelerated increase in life expectancy of the Chilean population implies new challenges. An aging population increases the likelihood of facing health challenges such as morbidity burden, chronic diseases, disability, and dependency [1]. Older people’s level of dependency and the requirement for long-term care stands out as a challenge, and yet, Chile is not prepared enough to deal with this complex situation [2]. Within the Chilean older population, the prevalence of dependency fluctuates around 15% to 20% [3,4,5], and this group is expected to increase [6].

Dependency entails higher levels of unmet needs and vice versa. Research suggests that unmet needs can result in a poorer quality of life, inefficient use of resources, and increased costs for health and social care services [7]. Hence, the coexistence of dependency and unmet needs indicates a condition of significant vulnerability and the environment’s failure to deliver necessary care.

Care is, in essence, a fulfillment of needs, but these include requirements that are unique to an individual; therefore, a comprehensive approach to the needs is required [8]. Such an approach relies on care systems, both formal and informal, which must be able to respond in a person-centered and dynamic way to the self-reported physical, environmental, social, and psychological needs [9,10,11,12]. To conduct a truly comprehensive needs assessment, it is crucial to consider self-perceived needs [13]. In this study, needs were measured according to the Camberwell Assessment of Need for the Elderly (CANE), a widely used instrument employed for over 20 years, primarily in European countries. The CANE is designed to measure a broad range of needs in older people, based on the principle that identifying a need means recognizing a problem plus an appropriate intervention to help or alleviate that need. A need is considered met when a mild, moderate, or severe problem receives an appropriate and potentially beneficial intervention [14]. Conversely, an unmet need indicates a serious problem requiring evaluation or an intervention that is either not being received or is being provided at an incorrect level or type of assistance. Through this approach, needs are classified into four domains: social, psychological, physical, and environmental [15]. Research using the CANE has shown that, among older people with dependency, self-perceived met needs are prevalent in physical and environmental dimensions, such as food, physical health, and home care [16]. This might be explained by the specificity of these needs, which makes them easier for caregivers to identify and address. In contrast, unmet needs predominantly concentrate on psychosocial dimensions, particularly the need for companionship, organizing daily activities, and managing psychological distress [16]. The difficulty for caregivers to accurately identify or meet these psychosocial needs jeopardizes the quality of life for older individuals [17].

Currently, only one study [18] has been conducted in Chile using the CANE. The study revealed that older Chilean people with dementia presented a higher average of caregiver-reported unmet needs compared to similar European studies. These findings are consistent with the lower levels of access to a variety of support for the needs of older dependent individuals and their caregivers in Chile. Furthermore, the same study identified caregiver characteristics, including the caregiver’s age, anxious symptomatology, perceived social support, and caregiver burden, as the sole predictors of unmet needs among people with dementia. Given this background, it is not clear whether these results can be replicated in the case of older adults who experience dependency but do not have cognitive impairment.

Regarding the community-dwelling older population without cognitive impairment in other countries, unmet needs tend to be positively related to worse physical functioning, greater frailty, a higher level of dependence on basic activities of daily living, a greater use of medications, and the presence of depressive symptomatology [11,12,19,20,21,22]. Conversely, an older person’s better quality of life and higher perceived social and emotional support are associated with a low frequency of unmet needs [12,22]. 

Interestingly, since studies employing the CANE in older people without cognitive impairment have not included caregivers in their samples, there is no literature about unmet needs and caregiver factors related to this group [11,12,20,21,22,23]. Consequently, little is known about the importance of the informal caregiver’s health status and characteristics to prevent unmet needs in their care recipients without cognitive decline. Also, since most of the studies on the topic come from Europe [16], where formal resources and service access are more extensive [10], data reflecting the Latin-American context are needed.

The aim of this study was to identify the unmet needs of dependent older individuals without cognitive impairment and their relationship to the personal factors of informal caregivers. Concerning the characteristics of older adults, it was hypothesized that (a) unmet needs would be more frequent in psychosocial areas, and (b) higher dependency and mental health issues would be associated with a greater presence of unmet needs in older adults. Regarding the relationship between caregiver characteristics and the unmet needs of the care recipient, although there was not enough previous evidence, it was expected that high levels of caregiver burden or mental health problems were related to a high number of older adults’ unmet needs.

## 2. Materials and Methods

### 2.1. Sample

This was a cross-sectional study with non-probability convenience sampling. It involved dyads of older persons with some degree of dependence in activities of daily living and their main informal caregivers, who agreed to answer a separate survey.

The inclusion criteria for older persons were as follows: they had to be aged 60 or over (which is the legal definition of an older person in Chile), have some degree of dependence (mild, moderate, or severe, according to the Barthel index), and have an informal caregiver (a family member or friend who performed the main caregiving activities). The dyad was excluded if the older person had cognitive impairment or if the caregiver was formal (i.e., paid).

Between April 2019 and February 2020, a total of 190 dyads were contacted. Some of these dyads did not meet the inclusion criteria for the study, while others declined to participate (see Figure 1). The final sample included 77 dyads, comprising 154 older people with dependency and their caregivers, most of whom lived in Santiago de Chile, the capital of the country.

### 2.2. Consent Procedure and Data Collection

Recruitment was carried out using different strategies. Advertisements for potential participants were shared through social networks. Also, health professionals at primary care facilities, outpatient centers, and caregivers’ associations helped identify potential participants and provided them with information about the research. To those who agreed to be contacted, detailed information about the study was given. People who accepted participation were surveyed at their homes. Informed consent was obtained from all participants, one for the older person and one for the caregiver. The surveys were carried out by psychologists trained on the instruments. Dependent older people were interviewed by a member of the team, while caregiver questionnaires were self-administered.

### 2.3. Instruments

#### 2.3.1. Administered to Older People

##### Camberwell Assessment of Need for the Elderly (CANE)

The CANE is a comprehensive instrument that assesses 24 areas of need, which in turn are classified into 4 dimensions (social, psychological, physical, and environmental), rated as no need, met need, or unmet need [15]. The CANE has good psychometric properties in terms of reliability (α = 0.99) and validity (correlated with the CAPE-BRS, r = 0.66; and the Barthel r = −0.53) [24]. As for the Spanish version, it presents good reliability properties (inter-observer reliability of 0.60 to 1 and test–retest coefficient of 0.65 to 1) [18]. It has a semantic adaptation for the Chilean population [18]. This scale was only applied to the older person.

##### Mini-Mental State Examination (MMSE)

It allows a general evaluation of the participant’s cognitive performance. The full version of the mini-mental test validated for Chile [25] was used, with a sensitivity of 93.6% (95% CI of 70.6–99.7%) and a specificity of 46.1% (95% CI of 34.7–57.8%). The cut-off score to indicate cognitive impairment is equal to or below 21 points out of a total of 30 points. In this sample, older people scored higher than 21 points.

##### Barthel Index of Activities of Daily Living

It evaluates the functionality of the respondent through the degree of independence reported in the performance of basic activities of daily living, such as feeding, dressing, bathing, and mobility. The cut-off scores of this instrument indicate that achieving 100 points represents independence. A score below 100 but equal to or greater than 60 points suggests mild dependency, a score equal to or greater than 40 points but less than 60 indicates moderate dependency, and any score below 40 signifies severe or total dependency. This instrument has shown good consistency with intra-observer Kappa indices of 0.47 and 1, inter-observer Kappa indices of 0.84 and 0.97, and a Cronbach’s alpha between 0.86 and 0.92 [26]. The Spanish version is available and does not require linguistic validation or adaptation [27]. Furthermore, it is widely used in primary care in Chile. 

#### 2.3.2. Administered to Older Persons and Their Caregivers

##### EuroQoL-5D: Visual Analog Scale (VAS)

The EQ-5D measures self-perceived health-related quality of life. The Visual Analog Scale (VAS) section was used, where the participant must rank their health from 0 to 100 points, where 0 is the “worst imaginable state of health” and 100 is the “best imaginable state of health”. It correlates significantly with the HAQ test (Stanford Health Assessment Questionnaire) (rs = −0.61 for the EQ-VAS) [28]. In Chile, the questionnaire has been adapted and used in several studies on quality of life [29,30]. 

##### Hospital Anxiety and Depression Scale (HADS)

This scale was developed to assess anxiety and depression symptoms in no clinical populations. It gives two independent total scores for anxiety and depression, both with a range of 0 to 21 points, where scores higher than 7 suggest the presence of symptomatology and equal or higher than 11 suggest the possibility of an anxious or depressive disorder [31]. This scale does not include items related to physical pathologies (which are not recommended when assessing anxiety and depression in older adults). The Spanish version has good psychometric properties of reliability and validity, with a test–retest coefficient of 0.85 and an internal consistency of α = 0.86 for both subscales; the 14 items show a clear two-factor structure, where all items showed a higher correlation (Pearson and Spearman) with their factors than with their opposites (the anxiety items presented correlations higher than 0.46, and the depression items were higher than 0.61) [32]. Furthermore, this instrument has been validated for similar populations and has been successfully employed in other studies in Chile [18,33].

##### Multidimensional Scale of Perceived Social Support

It is composed of 12 items that collect information on the social support perceived by individuals in two subscales: family/significant others and friends [34]. The Chilean validation [34] shows good reliability indexes (α = 0.86 for the total scale; α = 0.86 and α = 0.88 for the family/significant other and friends subscales, respectively). In the same study, the structure of the scale was evaluated based on exploratory and confirmatory factor analysis. The findings revealed that the two-factor model explains 59.2% of the variance, and the item correlations are greater than 0.59 and 0.81 for the family/significant other and friends subscales, respectively. 

#### 2.3.3. Self-Administered by Caregivers

##### Zarit Burden Interview

It measures the level of burden experienced by caregivers. This scale has been validated in Chile [35], obtaining an internal consistency value of α = 0.87, interobserver reliability (intraclass correlation coefficient: r = 0.86, 95% CI = 0.81–0.91), and test–retest stability reliability (Kappa index of 0.91, 95% CI = 0.68–0.99). The construct validity was evaluated based on convergent validity, where a Pearson correlation was obtained with a single burden indicator constructed by the authors (r = 0.67), and it also correlates with the ICD-10 Depression Survey (r = 0.7).

##### Revised Self-Efficacy for Caregiving Scale

It consists of three subscales: self-care and obtaining respite, controlling upsetting thoughts, and responding to disruptive behaviors [36]. The third subscale focuses specifically on disruptive behaviors commonly observed in older people with dementia, such as repeating the same questions. Since the present study excludes participants with a dementia diagnosis, this subscale was not utilized. The concurrent criterion validity of both remaining subscales was assessed. The “self-care and obtaining respite” subscale showed a positive correlation with social support, while the “controlling upsetting thoughts” subscale exhibited negative correlations with depression, anxiety, and anger [37]. While it has not undergone formal validation in Chile, the instrument has appropriate psychometric properties, including reliability and good construct validity. It has been used in various samples of caregivers [38,39], including Latino caregivers [40].

### 2.4. Statistical Analyses

All statistical analyses were carried out using IBM SPSS Statistics for Windows, Version 28. A descriptive analysis was performed to characterize the sample and its met and unmet needs. To determine older people’s and caregivers’ factors significantly associated with unmet needs, bivariate analyses were undertaken. A multiple regression analysis was carried out to determine predictors of older people’s unmet needs using a stepwise method. Independent variables considered were older people’s functionality, the number of medications taken, anxious and depressive symptomatology, perceived social support, and health-related quality of life. In addition, caregivers’ burden, anxious and depressive symptomatology, perceived social support, and self-efficacy were included. All analyses were conducted at a significance level of *p* < 0.05 and *p* < 0.01.

## 3. Results

### 3.1. Characterization of Dependent Older People

The mean age was 79.9 years (SD 9.9; range 62–101). Most participants were women (67.5%), some were widowed (45.5%), and many had only a few years of education (see Table 1). The socioeconomic level was measured according to the respondent’s report of their income, which, on average, was below the minimum wage. The majority (96%) of respondents reported having two or more diagnosed chronic health problems, including high blood pressure; diabetes; rheumatological diseases (arthritis, osteoarthritis, osteoporosis); chronic renal insufficiency; chronic respiratory diseases; sensory impairments such as visual limitations, hearing impairment, and deafness; as well as cancer, among others. All respondents took at least one medication, with a mean of 5.6 currently prescribed medications (SD 3.1; range 1–15). The most prevalent level of dependence was mild (74%), followed by moderate (18.2%) and severe dependence (7.8%). The mean observed for depressive symptomatology was 6.3 (SD of 4.0; range of 0–17), with 15.6% of the sample exceeding the cut-off point (11 or more points), indicating the possibility of a depressive disorder. For anxious symptomatology, the mean was 6.6 (SD if 4.8; range of 0–18), with 19.5% of the sample exceeding the cut-off point, suggesting the possibility of an anxiety disorder. 

### 3.2. Demographic and Clinical Characteristics of Caregivers

Higher age variability was observed (M = 55.7; SD if 14.3; range of 18–83), where most caregivers were young or middle-aged adults (62.3%) or older persons (37.7%). Most caregivers were sons/daughters (67.5%), followed by spouses (19.5%) of the dependent older person (see Table 2). Most caregivers were women (84.4%) who had complete secondary education or who held a technical/professional degree (73.2%). Their income was also variable, with an average slightly above the minimum wage. The main diagnoses for caregivers were high blood pressure (42.9%), diabetes (13%), and depression (13%), where comorbidity was also observed. Some participants reported having no diagnosis at all (23.4%). More than half of the caregivers presented moderate or intense burden (58.5%). In addition, 24.7% of them exceeded the cut-off score for a depressive disorder, and 32.5% did for anxiety disorder.

### 3.3. Older People’s Met and Unmet Needs

The average total needs were 9.4 (SD 3.2), of which met needs had a mean of 7.5 (SD of 2.2; range of 3–14) and unmet needs of 2 (SD of 2.3; range of 0–8). The most common met needs were “Food” (79.2%), “Physical health” (93.5%), and “Looking after home” (77.9%). The most frequent unmet needs were “Daily activities” (33.8%), “Company” (23.4%), and “Psychological distress” (24.7%) (see Table 3). Thus, the most common met needs belonged to the environmental or physical areas, while the most prevalent unmet needs were found in psychosocial areas.

### 3.4. Factors Associated with and Predictors of Unmet Needs

Greater functionality (rs = −0.53, *p* < 0.01), perceived social support (rs = −0.25, *p* < 0.05), and health-related quality of life (rs = −0.16, *p* < 0.05) were associated with a lower number of unmet needs. On the contrary, a positive relationship was observed between unmet needs and depressive (rs = 0.47, *p* < 0.0) and anxious symptomatology (rs = 0.35, *p* < 0.01). No significant correlations were obtained between unmet needs and characteristics of the informal caregiver (see Table 4).

Of all the predictors introduced in the regression model, only two proved to be significant: level of functionality (b = −0.38; *p* = 0.00) and anxious symptomatology (b = 0.33; *p* = 0.00), resulting in a model with a predictive capacity of 31% (see Table 5).

## 4. Discussion

To our knowledge, this is one of the first studies in Latin America to assess the needs of dependent older people through the CANE. This instrument was useful in capturing the self-perceived needs of older people with mild to moderate dependence and without cognitive impairment.

### 4.1. Unmet Needs of Older Dependent People in Chile 

The overall average of unmet needs in our sample (2.0) was higher than those reported in European studies with similar samples [16]. For instance, a study conducted in the Netherlands on self-perceived needs in older adults with joint pain and comorbidity [9] reported an average of 0.3 unmet needs, while a sample of frail older adults in primary care in the same country [10] had a mean of 0.5 unmet needs. Furthermore, consistent with previous literature, the mean number of unmet needs in our study was lower compared to older people with cognitive impairment [16,18]. 

We also found that most physical and environmental needs were met, while a great number of unmet needs were observed in the psychosocial dimensions. This pattern is consistent with previous evidence obtained from the CANE [16]. This can be explained by the fact that physical and environmental needs (such as food, physical health, and home care) are easier to identify and, in general, can be addressed by families themselves. In contrast, needs of a social and psychological nature run the risk of being overlooked by both families and formal care [16]. The former might normalize psychosocial problems, such as isolation, sadness, and anxiety, as part of the aging process [41]. In addition, informal caregivers (who are mostly burdened) may face difficulties or lack of preparation to meet psychosocial needs on their own. On the other side, the formal health care system may focus on dependency in old age only from a physical perspective, and consequently, psychosocial problems remain undervalued or underdiagnosed [42]. 

### 4.2. Factors Associated with Unmet Needs

The results revealed that perceived social support and health-related quality of life were negatively associated with the number of unmet needs. These findings are consistent with existing evidence suggesting that these variables can act as useful resources in addressing adverse situations such as physical and mental illness and loneliness [43,44,45]. Furthermore, they are positively correlated with other variables, such as life satisfaction [46,47]. Additionally, in our study, a positive relationship was observed between depressive and anxious symptomatology and unmet needs. This aligns with prior research, which consistently indicates a significant association between mental health variables and the presence of unmet needs [16].

### 4.3. Predictors of Unmet Needs

Unsurprisingly, the level of dependence was the main predictor of unmet needs. Regarding this, in the Chilean primary care system, there is a program for those with severe dependency [48] and another one for people who are self-sufficient or at risk of developing dependency [49]. Although day centers have been recently expanding, care services and public policies for older people with mild to moderate levels of dependence are still insufficient, and they are not available across some areas of the country. In consequence, older people who are neither totally self-sufficient nor severely dependent do not receive enough preventive and health promotion support, exposing them to a further increase in dependency. Something similar occurs with their caregivers, who are more likely to seek and receive help when they have already high levels of burden and develop physical and/or mental health problems.

Anxiety symptoms were found to be a significant predictor of the number of unmet needs. Furthermore, in this study, older people had a high level of anxious symptomatology, and a significant number of them even presented scores above the cut-off point, suggesting the presence of a clinical disorder. The relationship between unmet needs and anxiety may be reciprocal, i.e., an older person with symptoms of anxiety might present more unmet health needs, and if these are not met in time, the anxious condition may increase. This reality is complex for older people, considering that frailty and disability are risk factors for developing depression and anxiety [50,51].

In contrast to what was previously hypothesized, none of the caregivers’ aspects were associated with the older persons’ unmet needs. Since participants in this sample had mainly mild or moderate levels of dependency, caregivers’ issues might not have had a substantial impact on older persons’ lives. These results highlight the usefulness of employing a systematic approach to needs assessment that considers both the perspective of older individuals with dependence and the one of their primary caregivers. This approach enables a comprehensive evaluation, thus identifying opportunities to develop tailored interventions. In this study, as the predictive factors came only from the older persons, the resources should focus directly on them, conducting secondary and tertiary prevention to hinder the progression of dependence and addressing their mental health needs. To achieve this, older people’s coping strategies can be improved. For example, identifying and enhancing areas where they still maintain autonomy, relying on them as resources to stay as active as possible, or reinforcing their abilities to recognize their support networks and how to activate them. Such interventions can be cost-effective through innovations such as counseling via telemedicine [52].

On the other hand, caregivers in this sample were expected to have lower levels of caregiver burden, as previous research indicated that lower dependence levels in older people are associated with reduced caregiver burden [53]. Nevertheless, a large proportion of the caregivers in the sample showed moderate or intense burden, and more than a quarter of the caregivers were classified as possible cases of depression or anxiety. This may be indicative of limitations in mental health services and psychoeducation services for caregivers. In Chile, although there are some programs focused on caregivers, their coverage level is extremely low [54].

Overall, it is essential to develop adequate services for the dyad to avoid both a rapid decrease in the functionality of the older person and the development of mental health problems in the dyad, favoring the well-being of the family and the right to healthy aging [55].

On the other hand, the results of this study can help to understand how patterns of need manifest in older individuals with reduced functionality in societies where services are limited and care falls mainly on families. In other words, these results may be better reflecting the reality of low- or middle-income countries.

### 4.4. Limitations and Strengths

This study has limitations. It is a cross-sectional study, so causality cannot be determined. The sample size was small, reducing the power to detect significant relationships. The recruitment process was adversely affected by a tight project execution timeframe, the rigorous inclusion criteria, and the small number of older patients registered at the service where most of the participants were recruited. Also, the convenience sample might not be representative of the population of dependent older people in Chile. However, the recruitment was made using different strategies, giving some variability to the sample in terms of the services received and the needs presented. Furthermore, some of the tests that were employed are not formally validated in Chile, such as the CANE and the Revised Self-Efficacy for Caregiving Scale. However, they were applied because of their good psychometric properties in similar samples.

A strength of the present study is that it is one of the first to assess the needs of dependent older people in Latin America using a standardized measure. Also, previous studies that assessed the needs of older persons through the CANE did not explore how older people’s mental health (not cognitive impairment) was related to their unmet needs. The results of this study suggest the importance of considering such factors as predictors of unmet needs, particularly anxiety.

## 5. Conclusions

Most unmet needs of dependent older people were found in psychosocial areas. Also, a higher number of unmet needs were predicted by lower functionality and anxious symptomatology of the older person. Furthermore, caregivers in this sample presented high levels of burden and mental health symptomatology, indicating that they also required support.

Although Chilean health and social services for older people have grown within the last few years, there is still an important gap in addressing the psychosocial unmet needs of this group and those of their family caregivers, with a special emphasis on their mental health. This should include both strategies to promote psychosocial well-being and interventions to ameliorate mental health problems once they are present.

## Figures and Tables

**Figure 1 ijerph-20-06928-f001:**
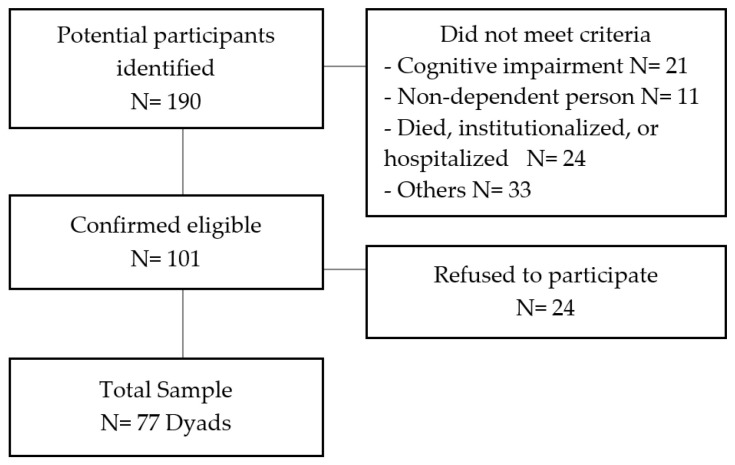
Flowchart of the recruitment process.

**Table 1 ijerph-20-06928-t001:** Sociodemographic and clinical characteristics of older adults with dependency (N = 77).

Sociodemographic Characteristics	Levels	Freq (%)/Mean (SD)
Age (years)	Mean (SD)	79.9 (9.9)
	60–73	22 (28.6)
	74–83	26 (33.8)
	84–101	29 (37.7)
Gender	Female	52 (67.5)
	Male	25 (32.5)
Marital status	Single	6 (7.8)
	Married/living with a partner	30 (39)
	Separated/divorced	6 (7.8)
	Widowed	35 (45.5)
Educational level	Primary education or less	21 (27.3)
	Incomplete secondary education	28 (36.4)
	Completed secondary education	18 (23.4)
	Professional or technician	10 (13)
Living with the caregiver	Yes	64 (83.1)
	No	13 (16.9)
Clinical characteristics		
Diagnosis	High blood pressure	61 (79.2)
	Diabetes	30 (39)
	Rheumatological diseases	29 (37.7)
	Chronic respiratory diseases	15 (19.5)
	Chronic renal insufficiency	10 (13)
	Cancer	9 (11.7)
	Sensory impairments	13 (16.9)
Level of dependency	Mean (SD)	70.4 (21.3)
	Mild	57 (74)
	Moderate	14 (18.2)
	Severe	6 (7.8)
Depressive symptomatology	Mean (SD)	6.3 (4)
	No symptoms	43 (55.8)
	Presents symptoms	22 (28.6)
	Possible disorder	12 (15.6)
Anxious symptomatology	Mean (SD)	6.6 (4.8)
	No symptoms	47 (61)
	Presents symptoms	15 (19.5)
	Possible disorder	15 (19.5)

**Table 2 ijerph-20-06928-t002:** Demographic and clinical characteristics of informal caregivers (N = 77).

Sociodemographic Characteristics	Levels	N (%)/Mean (SD)
Age (years)	Mean (SD)	55.7 (14.34)
	18–40	11 (14.3)
	41–60	37 (48)
	61–83	29 (37.7)
Gender	Female	65 (84.4)
	Male	12 (15.6)
Educational level	Incomplete high school education	20 (26)
	Completed high school education	23 (29)
	Professional or technician	34 (44.2)
Relationship with the older person	Partner/spouse	15 (19.5)
	Son/daughter	52 (67.5)
	Sibling	3 (3.9)
	Grandchild	6 (7.8)
	Mother	1 (1.3)
Clinical characteristics		
Diagnosis	Hypertension	33 (42.9)
	Diabetes	10 (13)
	Depression	10 (13)
	No diseases	18 (23.4)
Caregiver burden	Mean (SD)	51.3 (16.7)
	Absence of burden	32 (41.6)
	Moderate burden	16 (20.8)
	Intense burden	29 (37.7)
Depressive symptomatology	Mean (SD)	6.5 (4.4)
	No symptoms	47 (61)
	Presents symptoms	11 (14.3)
	Possible disorder	19 (24.7)
Anxious symptomatology	Mean (SD)	8 (4.9)
	No symptoms	33 (42.9)
	Presents symptoms	19 (24.7)
	Possible disorder	25 (32.5)

**Table 3 ijerph-20-06928-t003:** Frequency (%) of met and unmet needs of the dependent older adults.

Needs (N = 77)	Met (%)	Unmet (%)	Total ^1^ (%)
Accommodation	42.9	1.3	44.2
Looking after home	77.9	1.3	79.2
Food	79.2	1.3	80.5
Self-care	68.8	0.0	68.8
Caring for another	7.8	0.0	7.8
Daytime activities	18.2	33.8	51.9
Memory	7.8	2.6	10.4
Eyesight/hearing	70.1	19.5	89.6
Mobility	46.8	7.8	54.5
Continence	39.0	5.2	44.2
Physical health	93.5	5.2	98.7
Drugs	45.5	6.5	51.9
Psychotic symptoms	5.2	2.6	7.8
Psychological distress	26.0	24.7	50.6
Information	27.3	14.3	41.6
Deliberate self-harm	3.9	2.6	6.5
Accidental self-Harm	6.5	0.0	6.5
Abuse/neglect	1.3	1.3	2.6
Behavior	0.0	0.0	0.0
Alcohol	0.0	0.0	0.0
Company	11.7	23.4	35.1
Intimate relationships	10.4	14.3	24.7
Money	29.9	3.9	33.8
Social benefits	26.0	23.4	49.4

^1^ The percentage of total needs includes both met and unmet needs, excluding those rated as “no need” according to the classification of need in the Camberwell Assessment of Need for the Elderly (CANE).

**Table 4 ijerph-20-06928-t004:** Correlation between older people’s and caregivers’ factors and older persons’ unmet needs.

	Correlation	*p*
*Older Person Factors*		
Functionality	Rs = −0.53	0.00 **
Number of medications	Rs = 0.22	0.06
Health-related quality of life	Rs = −0.16	0.18
Social support	Rs = −0.25	0.02 *
Depressive symptomatology	Rs = 0.47	0.00 **
Anxious symptomatology	Rs = 0.35	0.00 **
*Caregiver Factors*		
Burden	Rs = 0.14	0.21
Social support	Rs = −0.04	0.71
Depressive symptomatology	Rs = 0.18	0.11
Anxious symptomatology	Rs = 0.14	0.23
Self-efficacy	Rs = −0.15	0.18

Note: * *p* < 0.05, ** *p* < 0.01.

**Table 5 ijerph-20-06928-t005:** Multiple regression analysis: predictors of unmet needs in the older dependent person without cognitive impairment.

Variable	Beta ^1^	*p* Value
Functionality (Barthel)	−0.38	0.00 **
Anxious symptoms (HADS-A)	0.33	0.00 **
R2	0.33	
Adjusted R2	0.31	
F	18.24	0.00 **

^1^ Note: Beta = standardized regression coefficient; ** *p* < 0.01.

## Data Availability

The data presented in this study are available on request from the corresponding author. Data are in the process of being archived and are not publicly available yet.

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
