# Peer review of "Predictors of Unmet Needs in Chilean Older People with Dependency: A Cross-Sectional Study"

_ijerph, 2023, doi:10.3390/ijerph20206928_

Round 1
Reviewer 1 Report
In my modest opinion, the manuscript entitled “Predictors of unmet needs in Chilean older people with dependency: A cross-sectional study” is a valuable and engaging contribution to the literature on aging. It is likely to be of interest to a broad audience of scholars, clinicians, educators, and students. There are, however, a few minor concerns that need to be addressed before the manuscript can be accepted for publication.
In the abstract and the introductory section, a clear link must be made between unmet needs and dependency. Most importantly, what are the likely consequences of the two variables coexisting in an older adult?
In the introductory section, hypotheses must be spelled out. Along with each hypothesis, a brief narrative of its rationale is to be presented. That is, each hypothesis must be justified.
In the introductory section, the variable needs must be discussed more thoroughly. The authors may consider different categories of needs (e.g., physiological, psychological, material, etc.) and their potential relationships. What are the implications of each of these different types of needs not being met? What is the available evidence? What are the coping mechanisms that help older adults deal with unmet needs?
In the method section, the authors state that older adults in Chile “have some degree of dependence (mild, moderate, or severe)”. What is the operational definition of each of these levels of dependency? How do the number and types of dependency contribute to each of these levels?
Some of the scales (e.g., Multidimensional Scale of Perceived Social Support and Mini-Mental State Examination) were validated on a Chilean population. What about the other scales? Were these scales translated? If so, how? If the translation was already available, was the translation validated for a Chilean population?
The sample is rather small. How did the author address this issue?
Before regression analyses, it is reasonable to present a table with correlation coefficients and coefficients of determination. If multicollinearity is present, the regression analysis cannot be computed.
In the discussion section, the authors may consider a broader view of their findings. To what extent do cultural differences limit the authors’ ability to generalize their findings to other populations of older adults? What do their findings add to the current state of the field (i.e., research on aging)? Most importantly, what are the coping mechanisms that older adults may be taught to use to deal with the sense of dependency they experience for needs that have remained unfulfilled? To what extent such coping mechanisms can mitigate a sense of dependency?
Minor editing of the English language is required
Reviewer 2 Report
I want to congratulate the authors on this very important study of a growing percentage of the world population - the elderly, especially those with dependencies- and the shortcomings they face, especially the burden they might create for those around them. I also appreciate the author's desire to analyse those in a middle position - not too well off to care for themselves and not too problematic that they need non-stop specialized help. I have a few questions/suggestions that, in my opinion, might clarify the manuscript.
1. Were information on the participant's living environment (urban/rural - small/large urban areas) collected? Are these relevant to the results?
2. Is the "family member - caregiver" position a result of financial issues, the fact the elderly's needs are not too demanding, the lack of specialized care centres or cultural elements? How much does each factor weigh? For example, if this is a cultural norm, maybe increasing the number of centres will not necessarily help.
3. If, as you point out, "None of the caregivers' aspects were associated with the older person's unmet needs" and "the formal health care system may focus on dependency in old age only from a physical perspective, and consequently psychosocial problems remain undervalued or underdiagnosed" how can the psychological unmet needs might be resolved?
